# The Effect of Extraction by Pressing at Different Temperatures on Sesame Oil Quality Characteristics

**DOI:** 10.3390/foods13101472

**Published:** 2024-05-10

**Authors:** Zahra Piravi-Vanak, Abdolah Dadazadeh, Sodeif Azadmard-Damirchi, Mohammadali Torbati, Fleming Martinez

**Affiliations:** 1Food, Halal, and Agricultural Products Research Group, Food Technology and Agricultural Products Research Center, Standard Research Institute (SRI), Karaj 31745, Iran; 2Department of Food Science and Technology, Faculty of Agriculture, University of Tabriz, Tabriz 51666, Iran; a.dadazadeh@tabrizu.ac.ir; 3Department of Food Science and Technology, Faculty of Nutrition, Tabriz University of Medical Sciences, Tabriz 15731, Iran; torbatim@tbzmed.ac.ir; 4Pharmaceutical-Physicochemical Research Group, Department of Pharmacy, Faculty of Science, The National University of Colombia, Bogotá 11001, Colombia; fmartinezr@unal.edu.co

**Keywords:** composition, oil extraction, oilseed, quality

## Abstract

Sesame oil has been widely used in the daily diet due to its high nutritional value. Sesame oil is extracted at industrial scales and also in small scale by cold pressing at different temperatures. In this research, sesame oil was extracted by pressing at four temperatures, namely, 30 (control sample), 60, 90 and 120 °C, to evaluate its effects on the quality of extracted oils. Oil extraction yields were increased from 38 to 51% by increasing the pressing temperature. The highest amount of peroxide and acid values were related to the oil extracted at 120 °C. Tocopherols and total phenol content were reduced by the increasing the pressing temperature, and the highest amounts of these bioactive components were related to the control sample. The results of the fatty acids profile showed that the composition of oils extracted at different temperatures did not differ significantly (*p* > 0.05). The results of the present study give a clear picture about the effects of different pressing temperatures on the sesame oil quality and extraction yield, and can be useful in the extraction unit optimization.

## 1. Introduction

Among the oilseeds, sesame (*Sesame indicum* L.) is one of the most important;it is economically important and is widely cultivated and distributed throughout the world [1]. This seed is considered as a significant oil (50–60%) and protein (18–25%) source. Sesame has many variations in color (from light cream to black), but white and brown varieties are more common and used for oil production [2,3].

Sesame seeds are primarily cultivated for oil extraction, and are also used for edible purposes, including the production of butter cake (sesame butter), halva, crackers and confectionery and bakery products [4]. The oil derived from sesame seeds has a mild smell and a pleasant taste, and for this reason, it is used as a natural oil for preparing salads and cooking oils or in the production of shortening, margarine, medicinal soaps, and insecticides. This oil has significant acceptance in different societies, due to its high nutritional value. Also, this oil is one of the more expensive oils among vegetable oils [5].

In recent years, consumer interest in virgin vegetable oils has increased worldwide, largely due to the growing evidence that these oils have beneficial nutritional and health effects [6]. One type of these oils are cold pressed oils [7]. Cold pressing does not mean that the oil does not tolerate processing at higher temperatures, but that high temperature is avoided to preserve the nutritional values and bioactive components without any qualitative changes in the nature of the produced oils [8]. Post-extraction treatments of these oils are usually performed by methods such as washing with water, sedimentation, filtering and centrifuging at temperatures lower than 45 °C without using any chemicals. Although the cold pressing method gives lower oil extraction yield compared to the other oil extraction methods, cold-extracted oil has natural and beneficial compounds such as tocopherols, sterols, carotenoids and phospholipids, which are well and more preserved [9].

Different pre-treatments such as moisture adjustment, roasting, ohmic heating and microwaving can be used to enhance the oil extraction yield and also to improve the extracted oil quality [10,11,12,13]. It is also possible to change the temperature during oil extraction from nuts and oilseeds by pressing and to use higher temperatures up to 200 °C [14]. In the oil extraction at the industrial scale and also in the retail market, different temperatures are used to extract oil from nuts and oilseeds [14]. The effects of sesame seed roasting at different temperatures and moisture contents before oil extraction by pressing were investigated on the obtained cake quality [15]. Also, the effects of different parameters including seed moisture content, pressing speed and restriction die were investigated in the sesame oil extraction by screw-pressing at low temperature (50 °C) [16]. However, to the best of our knowledge, there is no report on the effects of the application of different temperatures during oil extraction by pressing on the quality of the extracted sesame oils. Therefore, the purpose of this research was to determine the effects of higher temperatures on the oil extraction yields and also on the quality of extracted oils from sesame seeds.

## 2. Materials and Methods

### 2.1. Materials

Sesame seeds of the Darab variety (DV) and Tak Shakhe Naz variety (TV) with a moisture content of 8.0 ± 0.5% were obtained from the Seed and Plant Improvement Institute (Karaj, Iran). All the solvents and chemicals (acetic acid–chloroform, potassium iodine, starch, thiosulfate sodium, ethanol, potassium hydroxide, *n*-heptane, tertbutylmethylether, tetrahydrofuran, methanol, folin-Ciocalteu reagent, *n*-hexane, sodium carbonate, fatty acid methyl esters) used in this study were obtained from Sigma–Aldrich Chemie (Steinheim, Germany).

### 2.2. Oil Extraction

Sesame seeds (2 kg) were cleaned of all the impurities and then their oil was extracted at four different temperatures of 30 (control sample), 60, 90 and 120 °C by a screw press (Screw Press Model 85 mm, Kern Kraft, Neckarwestheim, Germany) according to the method described by Mazaheri et al. [10]. Nozzle size and press head were 12 mm and 8 mm, respectively. Rotation speed was about 50 rpm. Extracted oil was filtered to ensure that clean oil without seedcake residue was obtained.

### 2.3. Oil Extraction Yield

The percentage of oil extracted from sesame seeds pressed at different temperatures was determined based on the weight of the obtained oil after pressing and the weight of seeds used for oil extraction, as the following equation:$$Extraction\;yield\;\left(\%\right)=\frac{Extracted\;oil\;amount\;(g)}{Seed\;weight\;(g)}\times 100$$

### 2.4. Peroxide Value

Oil sample (5 g) was dissolved in the acetic acid–chloroform mixture (3:2 *v*/*v*) (30 mL) and a saturated solution of potassium iodine (0.5 mL) was then added and well mixed. The obtained mixture was kept in the dark for 1 min, and then water (30 mL) and starch solution (1 mL) were added and well mixed. The produced dark blue solution was titrated by thiosulfate sodium (0.01 N). The PV of a blank sample (without oil) was also determined and then the peroxide value (PV) of the oil sample was calculated according to the volume of thiosulfate sodium used by the following equation [17]:$$PV(\frac{meqO2}{kg\;oil})=\frac{\left(A-B\right)*N*1000}{oil\;weight\;(g)}$$
where A and B are the volumes of thiosulfate sodium used for titration of the oil sample and blank, respectively, and N is the normality of thiosulfate sodium.

### 2.5. Acid Value

The oil sample (20 g) was dissolved in the ethanol (50 mL)–chloroform (50 mL) mixture and 4 drops of phenolphthalein were added. Then, the mixture was titrated with potassium hydroxide (0.1 N) and the acid value (AV) was determined according to the used volume of potassium hydroxide using the below equation [17].
$$AV=\frac{V*N*56.4}{oil\;weight\;(g)}$$
where V is the volume of potassium hydroxide used for titration of the oil sample, and N is the normality of potassium hydroxide.

### 2.6. Determination of Steroid-Alike Compounds

Dehydrated sterol compounds were determined according to the IOC method [18]. The steroidal hydrocarbon was separated by column chromatography on silica gel (Silica gel 60) charged with silver nitrate, then it was identified by gas chromatography (DANI 1000, Milan, Italy) equipped with fused silica capillary column (0.25 mm i.d. by 25 m length) coated with 5%-phenylmethyl silicone phase, 0.25 μm film thickness). Cholesta-3,5-diene was used as an internal standard. Split injector was used with 1:15 flow divider. Injector and detector temperatures 300 and 320 °C, respectively. Oven programming temperatures was as follow initial 235 °C for 6 min and then rising at 2 °C/min up to 285 °C.

### 2.7. Tocopherols

The tocopherol compositions and contents of oil samples were determined by HPLC (Cecil Instruments, Cambridge, UK) based on the method described by Azadmard-Damirchi & Dutta [19]. The oil sample (10 mg) was mixed with *n*-heptane (1 mL). Then, the proper amount (about 10 μL) of the prepared mixture was used for injection and analysis. LiChroCART 250-4 Guard Column (Sigma-Aldrich, St. Louis, MA, USA) filled with LiChrosphere 100 (5-μm particle size) was used for tocopherol separations. The mobile phase was a mixture of *n*-heptane, tertbutylmethylether, tetrahydrofuran and methanol (79:20:0.98:0.02, *v*/*v*) at a flow rate of 1 mL/min. The detector was Varian 9070 fluorescence detector (Varian, Las Vegas, NV, USA) and used at 294 and 320 nm for excitation and emission, respectively. The retention times of pure tocopherols were used to detect the tocopherols in the analyzed oil samples. An external standard procedure was also used to calculate the concentrations of the tocopherols.

### 2.8. Total Phenol Content

The Folin-Ciocalteu reagent method described by Günç Ergönül et al. [20] was used to determine the total phenolic compounds (TPCs). Oil samples (2.5 g) were dissolved in *n*-hexane (2.5 mL) and in a mixture of methanol–water (2.5 mL, 80:20, *v*/*v*), and then centrifuged. The upper layer was separated, and the extraction step was done again. The extracts were combined and Folin-Ciocalteu reagent (2.5 mL) was added, and the mixture was mixed very well. Then, sodium carbonate saturated solution (5 mL) was added, and the total volume was made up to 50 mL, using distilled water. After 1 h, the absorbance was measured at 765 nm, using a spectrophotometer (Aquaris 1100, Cecil Instruments, Cambridge, UK). It was used to plot the standard curve that was prepared using the caffeic acid, and values were reported as mg caffeic acid/kg oil.

### 2.9. Fatty Acid Profile

For fatty acid analysis, fatty acid methyl esters (FAMEs) of the oil samples were prepared and then analyzed by gas chromatography (GC) (Agilent 7890 B, Agilent, Santa Clara, CA, USA) [19]. A flame ionization detector (FID) at 250 °C, split/splitless injector at 230 °C and BPX70 capillary column (50 m × 0.22 mm, 0.25 μm) were used in the GC for analysis. The carrier and make-up gases were helium and nitrogen, respectively. The temperature program for FAMEs analysis was stated from 158 °C for 5 min and then increased to 220 °C at a rate of 2 °C/min. FAMEs were detected using the retention time of standard FAMEs, and the peak areas were used to calculate the fatty acid percentage.

### 2.10. Statistical Analysis

Analysis of variance (ANOVA) was used for statistical evaluation, and Duncan’s multiple range test at a level of *p* < 0.01 was used to determine the significant differences between the means of obtained results, which were performed in triplicate. SPSS 18.0 (SPSS Inc., Chicago, IL, USA) was the statistical analysis software.

## 3. Results and Discussion

### 3.1. Oil Extraction

Oil extraction from oilseeds by pressing is a traditional method, which is used at industrial and also at retail scales. The obtained results showed that even if the press temperatures were as high as 120 °C, the temperature of oil, which comes out from the press, did not go higher than 53 °C (Table 1). This is an interesting result showing that it is possible to increase the temperature of the oil extraction process (from 30 to 120 °C) without a high increment in the temperature of the extracted oils (up to 48–53 °C). It has been reported that increasing the temperature of pressing during oil extraction from different types of nuts did not increase the temperature of the extracted oils due to the cooling effects produced from the continuous supply of raw materials to the barrel of the pressing system [14]. This can preserve the extracted oil quality, and an oxidation reaction would not occur, as happens in the higher temperatures [21,22].

Oil extraction yield is an important factor from economical points of view [23]. Oil extraction by cold pressing gives a lower extraction yield compared with hot pressing and solvent extraction. The results of this study showed that it is possible to increase the oil extraction yield by 10% (from 38 to 48% in DV and from 42 to 51% in TV) by increasing the temperature of the press from 30 to 120 °C (Table 1). However, the obtained results showed that extraction at 30 °C gives the lowest yield, and there were no significant differences in the oil extraction yields at temperatures of 60 and 90 °C (Table 1). It has been reported that sesame seeds give 37–63% oil, which is comparable with the obtained results in this study [7].

In addition to the pressing temperature, it has been reported that seed moisture content and pressing speed could also affect the oil extraction yield [16,23]. Higher and lower moisture contents could decrease the oil extraction yield, and there is an optimum moisture content for different seeds [10,16]. Also, increasing pressing speed from 20 to 60 rpm could decrease the oil extraction yield from sesame seeds [16].

At the industrial scale, oil is extracted from oilseeds with a high content of oil (higher than 20%), such as sesame seeds, by pressing, and the remaining oil in the produced cake from pressing is extracted by solvent, which is an economical procedure [24]. As retail oil extraction is small in scale compared with the industrial scale and does not produce a high amount of cake, there is no value in extracting the remaining oil in the produced cake by other methods. Therefore, at the retail scale, it is of high importance to extract the oil content of seeds as much as possible and reduce the remaining oil content in the cake without changing the quality and consumer acceptability of the extracted oils.

### 3.2. Peroxide Value

Peroxide value (PV) is one of the most important quality factors in the evaluation of edible oils and fats. PV shows the amount of primary oxidation product which is produced during the lipid oxidation due to several reasons, such as high degree of unsaturation, exposure to the air (oxygen), enzyme activity, low amounts of antioxidants, high prooxidant contents, poor oilseed quality and harsh storage conditions [25].

The obtained results showed that one of the important factors that can affect the PV level is the temperature during the oil pressing extraction. Increasing the temperature of pressing could increase the PV of the extracted oil samples (Figure 1). Also, there were differences in PV between the oils extracted from two sesame varieties, and oil extracted from DV had lower PV compared to TV (Figure 1). All the extracted oil samples had the acceptable PV according to the Codex Alimentarius standard, in which the PV has to be lower than 15 (mEq O_2_/kg oil) for virgin and cold-pressed vegetable oils [26]. It has been reported that increasing the temperature of the pressing barrel during almond oil extraction could enhance the PV of the extracted oils [14].

Higher temperature can increase the PV of oils due to increasing the oxidation reaction rate. The results showed that the quality of oil extracted from a cold press at lower temperature, for example 30 °C, could affect the PV of oil extracted at higher temperatures. The PV of oil extracted from DV had a low value at 30 °C, and therefore, the PV did not change significantly by increasing the temperature during the oil extraction (Figure 1).

### 3.3. Acid Value

Acid value (AV) shows the level of free fatty acids (FFAs) in the vegetable oils. FFAs are formed via heating in the presence of moisture or by the enzyme activity. The higher level of FFAs is sign of low-quality edible oil, which can be due to the poor quality of seeds (oil source), harsh processing condition, or unsuitable storage condition of vegetable oils [27]. Also, high level of FFAs can deteriorate vegetable oils quality and enhance its oxidation and also reduce the quality of foods prepared with these types of oils. Therefore, there is an upper acceptable limit for AV for vegetable oils. The AV is maximum 4 (mg KOH/g oil) for virgin and cold press vegetable oils such as sesame oil as established by international standards [26].

The obtained results showed that the extracted oils from both varieties of sesame seeds had AVs within the acceptable range (less than 4 mg KOH/g oil) (Figure 2). Oil extracted from TV had higher AVs compared with oil extracted from DV (Figure 2). It has been reported that variety can affect the composition and free fatty acid percentage of the sesame oil [28]. However, the AVs of extracted oils were increased by increasing the temperature of pressing. The increase in the AV was at a lower level during the oil extraction at temperatures lower than 90 °C, and the increase in the AV was more significant during the oil extraction at higher temperature (120 °C) (Figure 2). Generally, the obtained results are in agreement with previously published data, which has reported that the AV of oil extracted from unroasted sesame seeds was 1.03 (mg KOH/g oil) [29]. It has been reported that different pre-treatments before oil extraction by pressing can affect the free fatty acid content of sesame oil as well. Dehulling caused a higher increase in the free fatty acid content compared with cooking. However, roasting pre-treatment of sesame seeds could decrease the free fatty acid content of the extracted oil [28].

It is important to extract oils with low FFAs as much as possible to have a stable oil during further storage or in the prepared foods. Also, it has been reported that pretreatment of oilseeds by roasting or microwaves can inactivate lipase enzymes and prevent an increase in the AV of extracted oils by pressing [10]. Therefore, there is a need for further investigation into pretreating the oilseeds such as sesame by roasting or microwaving, and then extracting oil at different temperatures by pressing to monitor their effects on the extracted oil quality.

### 3.4. Steradienes

Phytosterols can be dehydrated and form steradienes, which can be formed during processing such as bleaching and deodorization or heat treatments [30]. Oil extracted from TV had a higher content of steradienes compared with DV (Table 2). Also, steradiene content was increased by increasing the pressing temperatures in both varieties. However, its content was below the acceptable level, which is ≤0.05 mg/kg for cold press oils [31]. Also, for extra virgin olive oil, ‘stigmastadiene’, which comprises the sum of stigmasta-3,5-diene and an unnamed isomer that elutes close to the -3,5-diene, has to be less than 0.15 mg/kg [32].

Ferrari et al. [33] also reported that higher treatment temperatures can increase the level of steradienes in vegetable oils. These compounds can be used as markers of vegetable oil quality and also for authentication purposes, particularly to distinguish virgin vegetable oils from the refined types [34].

### 3.5. Tocopherols

Tocopherols are bioactive components that are present in vegetable oils. They are important from the nutritional and technological points of view and play different roles, such as vitamin E and/or other antioxidants [35]. γ- and δ-tocopherols were detected in the extracted sesame oils, which were higher in the extracted oil from TV than that from DV (Table 3). Tocopherol content was decreased by increasing the temperature of pressing, but the decrease in the δ-tocopherol (25–50%) was higher than the decrease in the γ-tocopherol (10–20%) (Table 3). These results are in agreement with results obtained for oil extraction from pumpkin seeds at different temperatures by pressing [36]. It has been reported that tocopherols can be decomposed by the heating process, and during frying as well [37,38].

### 3.6. Total Phenol Content

Phenolic compounds can have effects on the stability, flavor and nutritional properties of vegetable oils [38,39]. Sesame seed oil is one of the edible oils with a high content of phenolic compounds, which makes it stable during storage and application as a cooking and frying agent [5]. Oil extracted from the TV had higher content of phenolic compounds compared with DV (Table 4). An increase in the temperature of pressing could decrease the phenolic compounds up to 10–15% in the extracted oils. Also, the decrease was enhanced in the temperatures higher than 60 °C (Table 4). These results are in agreement with results obtained for the total phenol content of oil extracted from pumpkin seeds at different temperatures by pressing, which showed that increasing the pressing temperature could decrease the phenolic content of extracted oil samples [40].

### 3.7. Fatty Acid Composition

Fatty acid composition is the most important factor in the nutritional value of fats and oils. It has important effects on the stability and technological properties of fats and oils as well. A higher level of saturated fatty acids has adverse effects on the nutritional properties of the fats and oils, but can increase the oxidative stability of these products. Therefore, a balanced ratio of saturated and unsaturated acids, particularly essential fatty acids, is necessary to achieve a suitable stability and optimal nutritional properties [41]. Sesame seed oil exhibited a high content of linoleic acid (18:2) as an omega-6 fatty acid, but it has a very low level of fatty linolenic acid (18:3), which is an omega-3 essential fatty acid (Table 5).

Oil extracted from DV had a higher content of C16:0 and C18:2, but lower content of C18:0 and C18:1 compared with oil extracted from TV (Table 5). Generally, different temperatures of pressing did not affect the fatty acid composition of extracted oils from both sesame varieties (Table 5). This result could be explained by the temperatures of the extracted oils during the process, which were lower than 53 °C, and also by the exposition time of oils to these temperatures, which were very short (i.e., less than 30 s). It should be mentioned that there were some fatty acids with very low amounts (<0.2) (C16:1, C17:1, C20:1 and C24:0) which were not included, as it could make the data and result presentation complex.

## 4. Conclusions

Sesame seed oil is extracted by cold pressing at the industrial and retail scales, due to the high and growing demands by the market. Different temperatures are used at the retail scale during extraction by pressing. The results of this study showed that the extraction yield, PV and AV of extracted oils increased by increasing the pressing temperature, although the PV and AV were still below the limits established by the Codex Alimentarius when the temperatures of the extracted oils were below 53 °C. There was a decrease in the total phenol and tocopherol contents with the increase in the pressing temperature. However, there were no significant changes in the fatty acid composition. Therefore, it can be concluded that heating can have a negative effect on the quality and nature of the oil obtained from the pressing procedures at temperatures above 60 °C.

## Figures and Tables

**Figure 1 foods-13-01472-f001:**
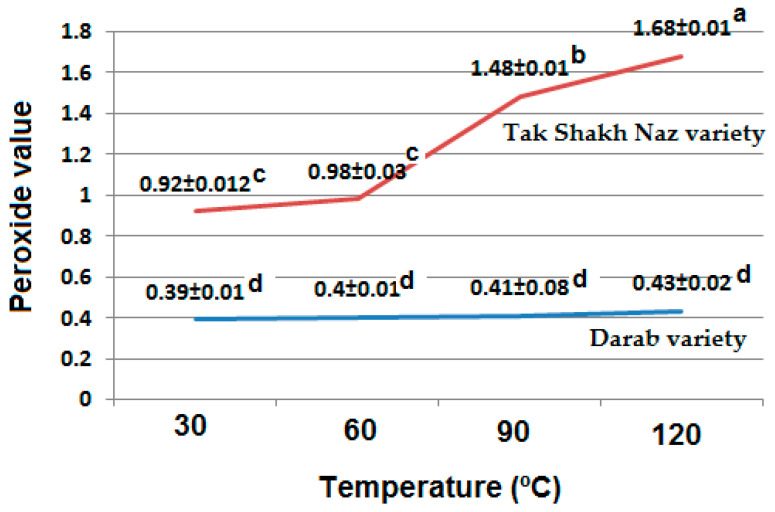
Effects of different pressing temperatures on peroxide value (mEq O_2_/kg oil) of the oil extracted from two sesame varieties. Different letters indicate a significant difference at the 5% level.

**Figure 2 foods-13-01472-f002:**
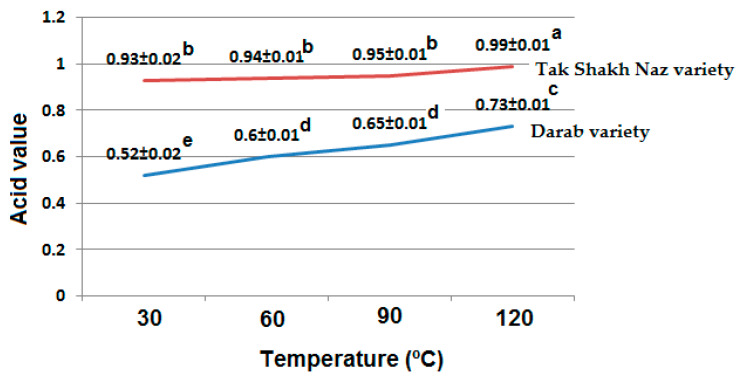
Effects of different pressing temperatures on acid value (mg KOH/g oil) of the oil extracted from two sesame varieties. Different letters indicate a significant difference at the 5% level.

**Table 1 foods-13-01472-t001:** Effects of different pressing temperatures on oil extraction yield (%) and temperatures of the oil extracted from two sesame varieties.

Temperature (°C)	Darab Variety	Tak Shakh Naz Variety	Extracted Oil Temperature (°C)
30	38 ^c^	42 ^c^	31–35 ^c^
60	44 ^b^	47 ^b^	37–40 ^b^
90	45 ^b^	48 ^b^	40–43 ^b^
120	48 ^a^	51 ^a^	48–53 ^a^

Different letters indicate a significant difference at the 5% level.

**Table 2 foods-13-01472-t002:** Effects of different pressing temperatures on steroid-like compound content (mg/kg) of the oil extracted from two sesame varieties.

Temperature (°C)	Darab Variety	Tak Shakh Naz Variety
30	0.005 ± 0.001 ^e^	0.010 ± 0.001 ^d^
60	0.008 ± 0.000 ^d^	0.030 ± 0.001 ^c^
90	0.011 ± 0.001 ^d^	0.038 ± 0.010 ^b^
120	0.025 ± 0.005 ^c^	0.045 ± 0.001 ^a^

Different letters indicate a significant difference at the 5% level.

**Table 3 foods-13-01472-t003:** Effects of different pressing temperatures on tocopherol content (mg/kg oil) of the oil extracted from two sesame varieties.

Variety	Press Temperature (°C)	γ-Tocopherol	δ-Tocopherol
Darab	30	610.8 ± 0.1 ^b^	24.5 ± 0.5 ^c^
	60	561.33 ± 1.8 ^c^	21.9 ± 1.2 ^d^
	90	548.3 ± 1.5 ^d^	18.6 ± 0.6 ^e^
	120	540.0 ± 0.1 ^cd^	18.5 ± 0.4 ^e^
Tak Shakh Naz	30	667.8 ± 0.01 ^a^	84.4 ± 0.4 ^a^
	60	662.9 ± 2.6 ^a^	80.7 ± 1.1 ^a^
	90	601.7 ± 2.7 ^b^	49.1 ± 0.5 ^b^
	120	534.7 ± 1.0 ^c^	45.2 ± 0.3 ^b^

Different letters indicate a significant difference at the 5% level in each column.

**Table 4 foods-13-01472-t004:** Effects of different pressing temperatures on total phenol content (mg/kg oil) of the oil extracted from two sesame varieties.

Temperature (°C)	Darab Variety	Tak Shakh Naz Variety
30	395.6 ± 0.1 ^c^	413.0 ± 0.1 ^a^
60	369.0 ± 1.3 ^e^	402.7 ± 0.1 ^b^
90	348.0 ± 0.1 ^f^	383.7 ± 2.0 ^d^
120	335.5 ± 0.1 ^g^	364.0 ± 3.8 ^e^

Different letters indicate a significant difference at the 5% level in each column.

**Table 5 foods-13-01472-t005:** Effects of different pressing temperatures on the fatty acid composition (%) of the oil extracted from two sesame varieties.

Variety	Press Temperature (°C)	C16:0	C18:0	C18:1	C18:2	C18:3
Darab	30	9.6 ± 0.2 ^a^	6.5 ± 0.1 ^b^	43.7 ± 1.4 ^b^	38.4 ± 1.6 ^a^	0.3 ± 0.01 ^a^
	60	9.4 ± 0.1 ^a^	6.4 ± 0.1 ^b^	43.3 ± 1.0 ^b^	39.1 ± 1.3 ^a^	0.3 ± 0.01 ^a^
	90	9.2 ± 0.1 ^a^	6.5 ± 0.2 ^b^	43.1 ± 1.7 ^b^	39.7 ± 2.0 ^a^	0.3 ± 0.01 ^a^
	120	8.9 ± 0.1 ^ab^	6.4 ± 0.1 ^b^	42.8 ± 1.1 ^b^	39.8 ± 1.0 ^a^	0.3 ± 0.01 ^a^
Tak Shakh Naz	30	8.3 ± 0.2 ^b^	6.6 ± 0.1 ^b^	48.2 ± 1.4 ^a^	35.4 ± 1.5 ^b^	0.2 ± 0.01 ^b^
	60	8.2 ± 0.2 ^b^	7.1 ± 0.2 ^a^	48.8 ± 1.1 ^a^	34.1 ± 1.0 ^b^	0.2 ± 0.01 ^b^
	90	8.2 ± 0.1 ^b^	6.7 ± 0.2 ^b^	48.0 ± 0.9 ^a^	35.2 ± 1.4 ^b^	0.2 ± 0.01 ^b^
	120	8.1 ± 0.3 ^b^	6.5 ± 0.1 ^b^	48.4 ± 1.7 ^a^	35.3 ± 1.8 ^b^	0.2 ± 0.01 ^b^

Different letters indicate a significant difference at the 5% level in each column.

## Data Availability

Detailed data are available upon request from the authors. The data are not publicly available due to privacy restrictions.

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
