# Peer review of "The Effect of Extraction by Pressing at Different Temperatures on Sesame Oil Quality Characteristics"

_foods, 2024, doi:10.3390/foods13101472_

Round 1

Reviewer 1 Report

Comments and Suggestions for Authors

The paper titled: “The effect of sesame oil extraction by pressing at different temperatures on its quality characteristics” represent a valid work, related to the evaluation of qualitative parameters of sesame oils obtained after the application of different temperatures. But at the same time does not represent a rich and innovative work.

I advise the authors to increase the discussion of the results, extending the statistical reports and improving the tables.

Comments on the Quality of English Language

Moderate editing of English language required

Author Response

Dear Editor and Dear Reviewers,

We would like to thank you very much for your valuable comments and suggestions which were very useful and helpful to improve the manuscript quality. We revised the manuscript extensively according to the comments and suggestions. Changes were made in red to be easily followed. Also, response to the comments and suggestions come as below:

Reviewer 1

Comments and Suggestions for Authors

The paper titled: “The effect of sesame oil extraction by pressing at different temperatures on its quality characteristics” represent a valid work, related to the evaluation of qualitative parameters of sesame oils obtained after the application of different temperatures. But at the same time does not represent a rich and innovative work.

I advise the authors to increase the discussion of the results, extending the statistical reports and improving the tables.

Author thank you very much for your valuable comments and suggestion. Manuscript was extensively revised as advised. Discussion part was extended and also the statistical works on Tables were rechecked as advised.

Reviewer 2 Report

Comments and Suggestions for Authors

In the manuscript “The effect of sesame oil extraction by pressing at different temperatures on its quality characteristics” authors investigated the effect of temperatures (30, 60, 90, 120°C) in the extraction of sesame oil from different varieties of sesame. They investigated the chemical composition, particularly fatty acid profiles, tocopherols, and total phenols. They also investigated some quality parameters, such as peroxide value and acid value. The manuscript is in the aim and scope of the Foods journal, and the authors did a good job. However, there are different issues. 

In the actual form, the manuscript is not reproducible, and the materials section must be improved. 

There are no graphs, which is unacceptable in a modern manuscript. I recommend at least a boxplot.

Furthermore, in L134, authors say that analyses were done in triplicate, but results in FAs composition are expressed with no standard deviation.

The English must be improved.

Here are my comments:

L41-L43: “In recent years, consumer interest in virgin vegetable oils has increased worldwide, largely due to the growing evidence that these oils have beneficial nutritional and health effects”. Can you add a reference about that?

L57-58: In the oil extraction at industrial scale and in the retail market oil extraction, different temperatures are used to extract oil from oilseeds such as sesame”. Can you add references about that? Which temperatures are used?

“However, to best of our knowledge, there is no valid and comprehensive report on the effects of application of different temperatures during oil extraction by press on the quality of extracted oils.” This statement is not entirely true. There are some papers about the extraction of sesame oil. Can you please add a proper introduction with references?

1. Rostami, M., Farzaneh, V., Boujmehrani, A., Mohammadi, M., & Bakhshabadi, H. (2014). Optimizing the extraction process of sesame seed's oil using response surface method on the industrial scale. Industrial crops and products, 58, 160-165.

2. Martínez, M. L., Bordón, M. G., Lallana, R. L., Ribotta, P. D., & Maestri, D. M. (2017). Optimization of sesame oil extraction by screw-pressing at low temperature. Food and Bioprocess Technology, 10, 1113-1121.

3. Kumari, K., Mudgal, V. D., Viswasrao, G., & Srivastava, H. (2016). Studies on the effect of ohmic heating on oil recovery and quality of sesame seeds. Journal of food science and technology, 53, 2009-2016.

4. Ji, J., Liu, Y., Shi, L., Wang, N., & Wang, X. (2019). Effect of roasting treatment on the chemical composition of sesame oil. Lwt, 101, 191-200.

Regarding materials and methods, please describe what you did more. Do not use references to explain what you did. Which is the method described by Mazaheri et al.? How was calculated the peroxide value according to the volume of thiosulfate sodium? Were the samples collected in the same year? From the same market producer? The solvents which grade? Which solvent did you use? Please be clear in the materials. As written, actually, the manuscript is not reproducible. 

I want the list of all the reagents and materials used in the “Materials” section.

L139-L140: “However, there is no scientific and reliable study on the effects of different  temperatures applied during the oil extraction by press in a retail scale on the oil extraction yield and also the quality of extracted oils”. This seems like an offensive statement. Please remove. 

L143-L144: “This is an interesting result showing that it is possible to increase temperature of oil extraction process without increment in the temperature of extracted oils.” Please rephrase properly. You have to comment the data with no adjectives. 

L145-L146: “This can preserve….temperatures”. Add references, if any.

L156-L162: Please improve the English of this part of the manuscript.

Regarding fatty acid composition, the analyses of fatty acids require standard. It is not clear which standard you have. Furthermore, the SUM of the fatty acids found in Table 7 is not 100%, even if expressed as a percentage. Why is there no standard deviation? 

Having 30 references in a manuscript may not be sufficient. Furthermore, the issue of self-citation needs addressing; one of the authors has self-cited in 4 out of the 30 references, constituting 13.3% of the total. It is recommended to expand the breadth of references by incorporating additional diverse sources to enhance the manuscript's credibility and breadth of literature review.

Comments on the Quality of English Language

The English must be improved.

Author Response

Dear Editor and Dear Reviewers,

We would like to thank you very much for your valuable comments and suggestions which were very useful and helpful to improve the manuscript quality. We revised the manuscript extensively according to the comments and suggestions. Changes were made in red to be easily followed. Also, response to the comments and suggestions come as below:

Reviewer 2

Comment1: In the manuscript “The effect of sesame oil extraction by pressing at different temperatures on its quality characteristics” authors investigated the effect of temperatures (30, 60, 90, 120°C) in the extraction of sesame oil from different varieties of sesame. They investigated the chemical composition, particularly fatty acid profiles, tocopherols, and total phenols. They also investigated some quality parameters, such as peroxide value and acid value. The manuscript is in the aim and scope of the Foods journal, and the authors did a good job. However, there are different issues. 

In the actual form, the manuscript is not reproducible, and the materials section must be improved. 

Response 1: Authors thank you very much for your valuable comments and suggestion. All comments were included in the revised version of the manuscript as advised.

There are no graphs, which is unacceptable in a modern manuscript. I recommend at least a boxplot.

Response 1: Peroxide value and acid value results were presented in Figures as advised.

Comment 2: Furthermore, in L134, authors say that analyses were done in triplicate, but results in FAs composition are expressed with no standard deviation.

Response 2: Standard deviations were included in the fatty acid table as advised.

Comment 3: The English must be improved.

Response 3: The English was rechecked and improved as advised.

Here are my comments:

Comment 4: L41-L43: “In recent years, consumer interest in virgin vegetable oils has increased worldwide, largely due to the growing evidence that these oils have beneficial nutritional and health effects”. Can you add a reference about that?

Response 4: This was from the below reference which was added in the text:

Rabiej-Kozioł, D.; Momot-Ruppert, M.; Stawicka, B.; Szydłowska-Czerniak, A. Health Benefits, Antioxidant Activity, and Sensory Attributes of Selected Cold-Pressed Oils. Molecules 2023, 28, 5484. https://doi.org/10.3390/molecules28145484

Comment 5: L57-58: In the oil extraction at industrial scale and in the retail market oil extraction, different temperatures are used to extract oil from oilseeds such as sesame”. Can you add references about that? Which temperatures are used?

Response 5: The effects of different temperatures in the cold press of some oil sources were investigated at 50, 100, 150 and 200 °C temperatures. The reference comes as below which were included in the manuscript text as well

Rabadan, A.; Pardo, J.E.; Gomez, R.; Alvarez-Oti, M. Influence of temperature in the extraction of nut oils by means of screw pressing. LWT–Food Sci. Technol. 2018, 93, 354–361.

Comment 6: “However, to best of our knowledge, there is no valid and comprehensive report on the effects of application of different temperatures during oil extraction by press on the quality of extracted oils.” This statement is not entirely true. There are some papers about the extraction of sesame oil. Can you please add a proper introduction with references?

  1. Rostami, M., Farzaneh, V., Boujmehrani, A., Mohammadi, M., & Bakhshabadi, H. (2014). Optimizing the extraction process of sesame seed's oil using response surface method on the industrial scale. Industrial crops and products, 58, 160-165.
  2. Martínez, M. L., Bordón, M. G., Lallana, R. L., Ribotta, P. D., & Maestri, D. M. (2017). Optimization of sesame oil extraction by screw-pressing at low temperature. Food and Bioprocess Technology, 10, 1113-1121.
  3. Kumari, K., Mudgal, V. D., Viswasrao, G., & Srivastava, H. (2016). Studies on the effect of ohmic heating on oil recovery and quality of sesame seeds. Journal of food science and technology, 53, 2009-2016.
  4. Ji, J., Liu, Y., Shi, L., Wang, N., & Wang, X. (2019). Effect of roasting treatment on the chemical composition of sesame oil. Lwt, 101, 191-200.

Response 6: Authors thank you very much for your valuable comment and providing very suitable papers to be included in the manuscript. All the mentioned references were used and included in the revised manuscript.

Comment 7: Regarding materials and methods, please describe what you did more. Do not use references to explain what you did. Which is the method described by Mazaheri et al.? How was calculated the peroxide value according to the volume of thiosulfate sodium? Were the samples collected in the same year? From the same market producer? The solvents which grade? Which solvent did you use? Please be clear in the materials. As written, actually, the manuscript is not reproducible. 

I want the list of all the reagents and materials used in the “Materials” section.

Response 7: All the solvents, chemicals and reagents were included in the material section and also all the methods were described in detail as advised.

Comment 8: L139-L140: “However, there is no scientific and reliable study on the effects of different  temperatures applied during the oil extraction by press in a retail scale on the oil extraction yield and also the quality of extracted oils”. This seems like an offensive statement. Please remove.

Response 8: The mentioned statement was deleted as advised.

Comment 9: L143-L144: “This is an interesting result showing that it is possible to increase temperature of oil extraction process without increment in the temperature of extracted oils.” Please rephrase properly. You have to comment the data with no adjectives. 

Response 9: This was revised and clarified by the obtained results and data.

Comment 10: L145-L146: “This can preserve….temperatures”. Add references, if any.

Response 10: Related references were added as advised.

Comment 11: L156-L162: Please improve the English of this part of the manuscript.

Response 11: The text was revised and clarified.

Comment 12: Regarding fatty acid composition, the analyses of fatty acids require standard. It is not clear which standard you have. Furthermore, the SUM of the fatty acids found in Table 7 is not 100%, even if expressed as a percentage. Why is there no standard deviation? 

Response 12: Fatty acid methyl esters standard was used for fatty acids determination, which was included in the text. Standard division was also added in the Table 7 as advised. There were some fatty acids with very low amount (<0.2) (for example C16:1, C17:1, C20:1 and C24:0) which were not included in the Table as it could make the Table complex. This issue was also explained in the text.

Comment 13: Having 30 references in a manuscript may not be sufficient. Furthermore, the issue of self-citation needs addressing; one of the authors has self-cited in 4 out of the 30 references, constituting 13.3% of the total. It is recommended to expand the breadth of references by incorporating additional diverse sources to enhance the manuscript's credibility and breadth of literature review.

Response 13: Thank you very much for this valuable comment. Introduction part, method section and discussion section were extended and new references were included in the revised manuscript as advised.

Round 2

Reviewer 2 Report

Comments and Suggestions for Authors

Authors answered to all my comments. The material and methods section and the references parts were improved as suggested.

Author Response

Dear Reviewers,

We would like to thank you very much for your valuable comments and suggestions which were very useful and helpful to improve the manuscript quality. We revised the manuscript extensively according to the comments and suggestions. Changes were made in red to be easily followed. Also, response to the comments and suggestions come as below:

Reviewer 1

Comment 1: In our opinion extraction is a method when were used organic solvent to obtain an oil. Here authors pressed the oil. So in the title and also in the text the word “extraction” should be delete.

Response 1: Authors thank you very much for this valuable comment. We agree with this comment, extraction is used to describe the extraction process by the solvent in the chemistry, but solvent and press are used for vegetable oil extraction and it is usual to describe the press method as extraction method. Here come some recent references which were used extraction term for pressing:

1-Effects of Extraction Processes on the Oxidative Stability, Bioactive Phytochemicals, and Antioxidant Activity of Crude Rice Bran Oil

Foods 2022, 11(8), 1143; https://doi.org/10.3390/foods11081143

2-The Effect of Different Extraction Methods on Extraction Yield, Physicochemical Properties, and Volatile Compounds from Field Muskmelon Seed Oil

Foods 2022, 11(5), 721; https://doi.org/10.3390/foods11050721

3-Influence of Pressure Extraction Systems on the Performance, Quality and Composition of Virgin Almond Oil and Defatted Flours

Foods 2021, 10(5), 1049; https://doi.org/10.3390/foods10051049 -

Comment 2: Line 173 should be “from 42 to 51% in TV”

Response 2: It was corrected as advised.

Comment 3: Line 179 – this part of the text is not connected with authors research

Response 3: This paragraph was deleted as advised.

Comment 4: All figures and tables - letters corresponding to statistical significance should be written as a superscript

Response 4: Letters corresponding to statistical significance were changed as advised.

Comment 5: Figure 1 there is no unit

Response 5: Units were highlighted in the Figure 1 and Figure 2.

Comment 6: In the text should be information about the moisture of the seeds. It is important for peroxide value.

Response 6: The moisture content was stated in the material section as advised.

Comment 7: Acid value should be some information about other publication comparable with that research

Response 7: The obtained results were compared with previously published data as advised.

Comment 8: Table 2 – there is no unit; each result and standard deviation should have the same decimal places

Response 8: Unit was added and standard deviations were revised as advised.

Comment 9: Line 324 should be „omega-3”, „omaga-6

Response 9: It was corrected as advised.
